



# Sources and atmospheric dynamics of organic aerosol in New Delhi, India: Insights from receptor modeling

Sahil Bhandari[1], Shahzad Gani[2], Kanan Patel[1], Dongyu S. Wang[1], Prashant Soni[3], Zainab Arub[3], Gazala Habib[3], Joshua S. Apte[2], and Lea Hildebrandt Ruiz[1]

[1]McKetta Department of Chemical Engineering, The University of Texas at Austin, Texas, USA
[2]Department of Civil, Architectural and Environmental Engineering, The University of Texas at Austin, Texas, USA
[3]Department of Civil Engineering, Indian Institute of Technology Delhi, New Delhi, India

**Correspondence:** Lea Hildebrandt Ruiz (lhr@che.utexas.edu), Joshua S. Apte (jsapte@utexas.edu)

**Abstract.**

Delhi, India, is the second most populated city in the world and routinely experiences some of the highest particulate matter concentrations of any megacity on the planet, posing acute challenges to public health (World Health Organization, 2018). However, the current understanding of the sources and dynamics of PM pollution in Delhi is limited. Measurements at the

5 Delhi Aerosol Supersite (DAS) provide a long-term chemical characterization of ambient submicron aerosol in Delhi, with near-continuous online measurements of aerosol composition. Here we report on source apportionment based on positive matrix factorization (PMF), conducted on 15 months of highly time-resolved speciated submicron non-refractory $PM_1$ ($NRPM_1$) between January 2017 and March 2018. We report on seasonal variability across four seasons of 2017 and interannual variability using data from the two winters and springs of 2017 and 2018. We show that a modified tracer-based organic component

analysis provides an opportunity for a real-time source apportionment approach for organics in Delhi. Thermodynamic modeling allows estimation of the importance of ventilation coefficient (VC) and temperature in controlling primary and secondary organic aerosol. We also find that primary aerosol dominates severe air pollution episodes.

## 1 Introduction

Exposure to fine particulate matter (PM) poses significant health risks, especially in densely populated areas (Pope and Dock-

15 ery, 2006; Apte et al., 2015). The Indian National Capital Region (Delhi NCR, India) is a rapidly growing urban agglomeration and encompasses the second-most populated city in the world with extremely high winter PM concentrations and frequent severe pollution episodes. According to a recent estimate, it is the world's most polluted megacity, on track to also become the world's most populated megacity by 2028 (World Health Organization, 2018; United Nations, 2018). However, the current understanding of the sources and dynamics of PM pollution in Delhi is limited (Pant et al., 2016b).

Delhi has a long history of studies focused on the quantity and composition of suspended particulate matter in Delhi (Mitra and Sharma, 2002). Several studies have found extremely high $PM_{10}$ concentrations (particulate matter smaller than 10 μm in diameter, $PM_{10}$~250–800 μg m$^{-3}$) and detected tracer compounds for vehicular emissions, biomass burning, and plastic burning. Large domestic use of fossil fuels and biofuels was found to correlate with especially high mass and number con-



centrations of PM observed in the evening and night. These studies captured various aspects of air quality patterns including diurnal variation, weekday–weekend effect, seasonal variation, interannual variation as well as correlations of total particle number and mass concentrations with gas phase species such as $SO_2$ and $NO_2$ (Sharma et al., 2003; Mönkkönen et al., 2004, 2005a, b). Other studies have discussed the speciation of mass in smaller particles (Chowdhury et al., 2007; Srivastava et al.,

2008; Tiwari et al., 2009). Recent studies such as Pant et al. (2015) have emphasized $PM_{2.5}$, and consistently report high winter concentrations of particulate chloride and nitrate, and high sulfate concentrations across both winter and summer seasons. They attribute chloride to sources such as coal, biomass burning, and waste burning due to the presence of molecular markers corresponding to these sources. They also attribute higher winter concentrations to condensation of semivolatile ammonium nitrate and ammonium chloride during low-temperature conditions, weaker wind speeds, and shallow atmospheric boundary

layer in the winter season (Pant et al., 2015, 2016a).

A recent review of receptor modeling studies with a focus on Delhi shows that most PM-based source apportionment studies have attributed Delhi's pollution to vehicular traffic, fossil fuel combustion and road dust (Pant and Harrison, 2012). However, the differences in particle size cutoff in different studies and techniques used made it difficult to compare results (Pant and Harrison, 2012). Previous receptor modeling studies have principally relied on a small number of daily or multi-day filter-

based samples collected over temporally restricted sampling periods, thereby limiting the possible application of factor analysis techniques such as positive matrix factorization (PMF). Further, despite Delhi being a continental site, multiple studies attribute significant portions of finer fractions of PM to a sea-salt origin (Sharma et al., 2014; Sharma and Mandal, 2017). Comparatively fewer studies have reported on $PM_1$ composition. One such study attributes winter (December) and spring (March) chloride peaks to be of non-sea-salt origin (Jaiprakash et al., 2017). They also focus on $PM_1$ composition and source apportionment in

Delhi and use HYSPLIT to point to potential chloride sources northwest of the Indian Institute of Technology (IIT) Delhi, such as industries of salt and metal processing and thermal power plants. As for organics, previous studies are mostly limited to the quantification of organic carbon, elemental carbon, polycyclic aromatic hydrocarbons, and water-soluble organic compounds (Sharma and Mandal, 2017; Singh et al., 2011; Pant et al., 2015).

The Delhi Aerosol Supersite (DAS) campaign provides a long-term chemical characterization of ambient submicron aerosol

in Delhi, with near-continuous online measurements of aerosol composition. Here we report on source apportionment conducted on 15 months of highly time-resolved speciated non-refractory submicron aerosols ($NRPM_1$), including organics, chloride, ammonium, sulfate, and nitrate. We use mass spectrometer data from an aerosol chemical speciation monitor (ACSM) in the PMF receptor model at a time resolution of 5–6 minutes. This work improves upon the low time resolution of source apportionment utilizing time-integrated filter-based measurements previously employed in Delhi. Over the campaign, organics

accounted for 53% of the submicron mass, followed by inorganics (36%, of which sulfate, nitrate, ammonium, and chloride contributed 13%, 8%, 9%, and 6% respectively) and black carbon (BC) (10%) (Gani et al., 2018). Table 1 provides a summary of key DAS bulk measurements.

In this manuscript, we focus on organic aerosol and report seasonal variability for four seasons and interannual variability using data from the two winters and springs of 2017 and 2018.





## 2    Methods

As a part of the DAS campaign, an Aerosol Chemical Speciation Monitor (ACSM, Aerodyne Research, Billerica, MA) was operated at ~0.1 liters per minute (LPM) at ~1-minute time resolution in a temperature-controlled laboratory on the top floor of a four-story building at IIT Delhi. Additionally, BC, ultraviolet absorbing particulate matter (UVPM) and their difference

ΔC were measured using a seven-wavelength aethalometer operated at 1 LPM flowrate and 1-minute time resolution (Magee Scientific Model AE33, Berkeley, CA) (Drinovec et al., 2015). These instruments were on separate sampling lines, both of which had a $PM_{2.5}$ cyclone followed by a water trap and a Nafion membrane diffusion dryer (Magee Scientific sample stream dryer, Berkeley, CA). For the ACSM, the scan speed was set at 200 ms amu$^{-1}$ and pause setting at 125 for a sampling time of 64 s. The ACSM measured mass from mass-to-charge ratio (*m/z*) *m/z* 10 to *m/z* 140. However, all analysis was restricted to

*m/z* 120 due to smaller mass and large uncertainties associated with data collected at the higher *m/z*s (Ng et al., 2011b). All essential calibrations such as flow rate calibration, ionizer tuning, quadrupole resolution adjustment, adjustment of multiplier voltage, *m/z* calibration, calibrations for measuring the response factor of nitrate, and relative ionization efficiencies of ammonium and sulfate were performed as recommended. Based on identified issues in the usual jump mode ionization efficiency calibrations, additional single scan mode calibrations for estimating ionization efficiencies were also conducted. For data pro-

cessing, airbeam corrections and the default relative ion transmission corrections were applied. Full details on the sampling site, instrument setup, operating procedures, and calibrations are described in a separate publication (Gani et al., 2018).

### 2.1    Study design

We collected the data used in this manuscript from January 2017 to March 2018. Since the ACSM measures mass spectra at every time point, we obtained a two-dimensional matrix of time points (rows) and mass spectral contributions (columns).

We conducted separate PMF analysis for each season, with our data categorized into the six distinct seasons over these 15 months (Table 2) (Indian National Science Academy, 2018). We used the dataset obtained by averaging every five consecutive measurements for the seasonal PMF runs. Autumn (mid-September to November) is not included in our core analysis due to the unavailability of ACSM data for that period.

We deployed the PMF2 program to resolve factors from the 2D matrices (Paatero and Tapper, 1994). The Igor PMF Eval-

uation Tool (PET) was used to conduct PMF2 analysis on this dataset and interpret its results (Ulbrich et al., 2009). Further details on the statistical basis of this method are available elsewhere (Ulbrich et al., 2009; Zhang et al., 2011, and references therein). Briefly, PMF is a bilinear unmixing model that performs deconvolution of mass spectra (MS) into the summation of products of positively constrained mass spectral profiles and their corresponding time series, under the assumption that the mass spectral profiles remain constant in time. The iterative PMF technique does not make any assumptions for source or time

profiles. In the process, the model minimizes the weighted least squares error (sum of squares of model error normalized to measurement error), or the summation of squares of scaled residuals of the fit at each *m/z* and each time point.

We used two alternative approaches for conducting PMF. In one approach, only organic spectral data at a specific set of *m/z*s between *m/z* 12 and *m/z* 120 were selected. This approach is the most commonly used approach, and the reasons for



the selection have been described previously (Zhang et al., 2005). In this approach, the concentrations of inorganic species measured by the ACSM were used only as external tracers for interpreting organic PMF factors. However, this technique provides limited information on the extent of the relationship between organic and inorganic species (Sun et al., 2012). We pursued a second approach in which we conducted PMF analysis for organic plus inorganic MS. The inorganic $m/z$s selected

represent the underived $m/z$s for each species such that spectral contributions at other $m/z$s can be explained completely by data at these $m/z$s (Jimenez group, 2018; Sun et al., 2012). Thus, we have conducted 12 PMF runs in total. We refer to the organic-MS-based PMF analysis results as "organic-only PMF" and combined organic-inorganic PMF analysis results as "combined organic-inorganic PMF" results in the manuscript.

    Within the PET tool, we removed spikes (Zhang et al., 2005) from the dataset and down-weighted selected weak and bad

$m/z$s with a low signal to noise ratio. Further, we used the default fragmentation table, and as a result, higher weight was given to mass spectral contribution at $m/z$ 44 with which data at $m/z$s 16, 17, and 18 are proportionally related. Accordingly, we down-weighted contributions at these $m/z$s. We readjusted the results from PMF analysis to account for underestimation of factor mass based on the selected $m/z$s only. To account for particle losses, we applied transmission and collection efficiencies after conducting PMF analysis (Gani et al., 2018).

## 2.2    Factor selection

We conducted PMF runs for one to six factors and explored the solution space using the tools FPEAK and SEEDs. We selected an initial number of p factors based on uncentered correlation coefficients with the factors at the selection p−1. Other criteria employed include improvement in the solution's ability to explain residual structure with the addition of a factor and ensuring the 25–75[th] percentiles of scaled residuals for all $m/z$s are between ±3. Changing SEEDs value initializes the PMF algorithm

with different pseudorandom starts. Changing FPEAK value allows exploring rotations of solutions of a given number of factors. Primarily, we gauged the effect of the changing FPEAK and SEEDs using changes in the fraction of variance explained by different factors, correlations of factors' MS with reference MS, and correlations of factors' time series with the time series of external tracers. Differences between plausible factor solutions in the FPEAK-SEED 2D space are also representative of the uncertainty of the final selected solution for each scenario in Table 2 (Ulbrich et al., 2009). We observed unreasonable

MS, weak time series correlations, or rotational ambiguity on changing FPEAK and/or SEED from the default selection of FPEAK=0 and SEED=0. We, therefore, used these default parameter values in our core analysis. Details on factor selection for each PMF run can be found in Supplementary Information (Sect. S1). The average mass spectral profiles developed by Ng et al. (2011a) based on previous ACSM and Aerosol Mass Spectrometer (AMS) research are used as reference mass spectra, and the one with the highest correlation with the mass spectrum of the PMF generated factor (generally, Pearson R $\geq$0.9) is

used for the terminology of the obtained factor (Ng et al., 2011a). This comparison allows the separation of factors based on their sources since MS of different factors are characterized by different spectral signature peaks (Zhang et al., 2011). For example, hydrocarbon-like organic aerosol (HOA) is a proxy for fresh traffic and combustion emissions and shows prominent peaks at $m/z$ 55 and 57 and a higher fractional organic signal at $m/z$ 43 than $m/z$ 44. When available, we used carbon monoxide (CO) measured about 11 km (~7 miles) from our site at a fixed monitoring location RK Puram maintained by the Central



Pollution Control Board (CPCB), Government of India as an external traffic and combustion tracer. Further, we also utilized the co-located aethalometer measurements of BC (880 nm) as a traffic tracer. For biomass burning, we used two tracers: (i) $\Delta C$, defined as the difference between UVPM (370 nm) and BC detected by the aethalometer (Wang et al., 2011; Olson et al., 2015; Tian et al., 2019) and (ii) the biomass burning component of black carbon, $BC_{BB}$, estimated using the model of Sandradewi
et al. (2008).

## 3   Results and discussion

In this manuscript, we focus on the components of organic aerosol obtained using organic-only PMF and combined organic-inorganic PMF. We report average seasonal concentrations of organic-only PMF factors in Table 3 and the organic component of the organic-inorganic combined PMF factors in Table 4. Our results show that the mass spectral profiles of organic-only
PMF factors are consistent with reference profiles. In five of the six seasonal organic-only PMF runs, we obtained only two factors—a mixed hydrocarbon-like organic aerosol (HOA)-biomass burning organic aerosol (BBOA) factor, hereafter referred to as primary organic aerosol (POA), and an oxidized organic aerosol (OOA) factor. PMF separated HOA and BBOA factors only in spring 2018, and POA MS for this season was calculated by adding these two factors, weighted by their respective time series contributions. Combined organic-inorganic PMF runs further separate the OOA factor but not the POA factor. Advanced
factor analysis including ME-2 will be explored in a forthcoming publication to separate the POA factor.

Figure 1a presents the mean of the seasonal organic-only PMF POA MS averaged over the entire campaign and the reference profile for HOA. The behavior of this POA factor is in line with the reference HOA factor (also, Figs. S1a–f, S2a–g, R>0.9), as suggested by the dominance of hydrocarbon signatures in the spectrum in the series $C_nH_{2n-1}^+$ and $C_nH_{2n+1}^+$ (Ng et al., 2010). At the same time, the fractional contributions of the POA factor at $m/z$s 29, 60, 73, and 115 are higher compared to the reference
HOA factor. These $m/z$s have higher contributions from biomass burning emissions than traffic-related and other combustion emissions (He et al., 2010; Crippa et al., 2014; Bertrand et al., 2017)—comparison with the reference BBOA profile in Fig. S3 points to the influence of biomass burning on the primary organic aerosol factor. As expected, POA tracers, CO and BC, correlate more strongly with the POA factor than with the OOA factor (Fig. S4a–f).

Figure 1b presents the mean of the seasonal organic-only PMF OOA MS, averaged over the entire campaign and the reference
profile for OOA. OOA is principally associated with secondary organic aerosol (SOA) (Zhang et al., 2011). Mass spectra of the OOA factors in organic-only PMF correlate strongly with the reference OOA factor (R>0.95) (Figs. S1a–f, S5a–f). Time series of the OOA factors correlate stronger with secondary inorganic species, such as nitrate and sulfate, compared to their correlations with the POA factor time series (Figs. S4a–f). Figure 2 shows the time series of the organic-only POA and OOA factors for the measurement period. The interplay of sources, meteorology, and reaction chemistry results in a sharp variation
of PMF factor concentrations across seasons. While POA and OOA concentrations are similar in the colder months, OOA is the more abundant component in the warmer months.

The combined organic-inorganic PMF analysis yielded three or four factors. One factor, POA, is predominantly composed of primary organics (accounting for 80–95% of the total factor mass), while an additional two or three factors emerge as a





combination of inorganics and oxidized organics—namely, ammonium sulfate (AS) mixed with OOA (ASOOA), ammonium nitrate (AN) mixed with OOA (ANOOA) and ammonium chloride (AC) mixed with OA (ACOOA). The time series correlations of the organic-inorganic combined PMF factors with external tracers are shown in Fig. S6a–f. We refer to the organic mass in these mixed factors as AS-OOA, AN-OOA, and AC-OOA respectively. This organic mass accounts for 19–59%, 21–56%,

and 17–23% of the total mass of these mixed factors, respectively. The mass spectrum of the POA from organic-inorganic combined PMF analysis correlates most strongly with the reference HOA and/or BBOA (R>0.9). AS-OOA and AN-OOA correlate most strongly with the reference LVOOA profile (R>0.95, Fig. S7a–f). It is therefore not surprising, as shown in Fig. 3, that the behavior of POA and OOA in combined organic-inorganic PMF is very similar to that of organic-only POA and OOA respectively. Based on the slope of the time series correlations, the combined organic-inorganic PMF estimates 8% more

POA and about 12% less OOA than organic-only PMF, with higher disagreement in warmer months. These differences may be due to a relatively higher $m/z$ 44 fraction in the POA from combined organic-inorganic PMF analysis than the POA from organic-only PMF analysis (Figs. S8–S13a). Overall, the MS and time series obtained by combining AS-OOA, AN-OOA, and AC-OOA are very similar to the organic-only PMF OOA MS and time series. The mass spectra and time series of the POA and OOA components obtained using combined organic-inorganic PMF factors and organic-only factors are consistently strongly

correlated in each season, as shown in Figs. S8–S13a–c.

In Sect. 3.1, we discuss the mass spectral profiles and diurnal time series patterns of POA across seasons. In Sect. 3.2, we discuss the mass spectral profiles and diurnal time series patterns of oxidized organic aerosol from the seasonal organic-only PMF analysis as well as the combined organic-inorganic PMF analysis. In Sect. 3.3, we contrast the importance of primary versus secondary organic aerosol. We also compare the full PMF results with the tracer-based organic component results.

In Sect. 3.4, we investigate interannual variability across the winters and springs of 2017 and 2018 and shed light on the association of PMF factor concentrations with ventilation-related variables —wind speed, planetary boundary layer height (PBLH), and ventilation coefficient (VC = PBLH × wind speed).

## 3.1   Primary organic aerosol (POA)

The analysis in this section focuses on PMF factors from the organic-only PMF analysis; as suggested by the mass spectral

and time series correlations, the behavior of POA in combined organic-inorganic PMF is very similar to the organic-only POA in each season (Fig. 3, Figs. S8–S13a, c) and is therefore not discussed here. The factors representing primary organic aerosol have consistently high correlations with hydrocarbon-like organic aerosol and show a varying influence of biomass burning organic aerosol (Fig. S1a–f). To assess the seasonal variability of this biomass burning influence on the mass spectra, Fig. S2a–g compares the mass spectra of the POA factors with reference HOA and BBOA profiles. The mixed POA profiles observed in

Delhi have fragments at $m/z$ 60 and 73, tracers of biomass burning, several times the HOA reference profile average and within 15% of the reference BBOA profile average in winter 2017 and 2018 (Fig. S2a,e). Even in the other three seasons with a mixed POA factor, $m/z$ 60 and 73 have contributions higher than the reference HOA profile average by more than a standard deviation (Fig. S2b–d). This behavior points to the mixing of biomass burning in the POA factor. In spring 2018, the separated HOA and BBOA are in line with their respective reference profiles (Fig. S2f–g). The mixing of HOA and BBOA factors in PMF has been





observed in previous studies as well (Aiken et al., 2009; Elser et al., 2016; Al-Naiema et al., 2018). The deployment of other factorization techniques including the Multilinear Engine (ME-2) algorithm for constraining the presence of POA sources such as cooking and biomass burning is the subject of ongoing work and a forthcoming publication.

The organic-only PMF-based POA diurnal time series patterns exhibit high diurnal, interseasonal, and interannual variability
and are influenced by episodic pollution episodes (Table 3, Fig. 4). Peaks in the POA diurnal pattern occur early in the morning and late at night, corresponding to periods of higher traffic. Winter peak POA concentrations are ~2× peaks in spring, ~3–5× peaks in summer, and ~6–7× peaks in monsoon. Additionally, winter peak POA concentrations are ~8–10× winter minima, and this peak-to-minimum ratio dampens in warmer months—decreasing to ~5–6× in summer. The larger nighttime (1800–2200 hours) and smaller daytime (0600–0900 hours) POA peaks are likely associated with nighttime emissions being trapped by
the decreasing PBLH (and increasing PBLH at daytime) (Figs. 4, S14) and minimal photochemical conversion of POA to SOA in the evening, confirmed by the lower OOA, AN-OOA, and AS-OOA concentrations in the evening (Sect. 3.2). Despite decreasing PBLH and ventilation coefficient at night (2100–0300 hours) (Fig. S14), POA concentrations are decreasing (Fig. 4) and are likely a consequence of decreasing emissions at nighttime and into the early morning of the next day. Additionally, the lower temperatures in the mornings and evenings of winter and spring seasons (Figs. 4, S14) also play a role in generating POA
peaks at these times: 87% of the 95[th] percentile episodes for POA occur between 10–20°C (Fig. S15e). Finally, the differences in POA minima across seasons is much smaller than the differences in peaks—POA minimum decreases from ~10–12 μg m$^{-3}$ in winter to ~4 μg m$^{-3}$ in monsoon, and all seasons seem to approach similar baseline concentrations in the afternoon. According to our analysis using the volatility basis set (VBS) (Donahue et al., 2006), differences in observed concentrations in summer and winter can be explained by thermodynamics (equilibrium partitioning) and meteorology (PBLH, VC) alone,
suggesting that sources of POA in Delhi may be similar in all seasons. The detailed methodology and results of the VBS analysis are presented in Supplementary Information (Sect. S2-VBS Application).

With regard to interannual variability, we observe notable consistency between daytime POA profiles in winter and spring of 2017 and 2018 (0900–1700 hours). However, at nighttime and early morning, POA concentrations have increased in both seasons in 2018 by as much as ~30 μg m$^{-3}$. It is well known that the median is robust against extremes and the arithmetic
mean (AM) is not. Thus, the relative difference between seasonally averaged mean and median is a qualitative measure of the influence of pollution episodes. Additionally, the magnitude of standard deviation (SD) relative to the arithmetic mean and the magnitude of geometric standard deviation (GSD) relative to the geometric mean (GM) are also evidence of episodic behavior. Indeed, based on these metrics of gauging the influence of episodes, POA is influenced by episodic events in the early morning hours in all seasons (Table 3, Fig. 4). These episodes increase mean POA concentrations over the corresponding median by as
much as 20 μg m$^{-3}$, with the largest episodic events occurring in winters. In relative terms, the largest difference is in summer, when mean POA exceeds the median by as much as 55%. Similarly, though seasonal AM and GM for POA are generally smaller than AM and GM for OOA, their corresponding GSD and SD are larger. This difference is particularly stark in colder months, with POA SD almost twice the OOA SD, despite similar AM. While ubiquitous temporally varying sources such as traffic and cooking are important contributors to the overall POA diurnal patterns, they have stable patterns within seasons
associated with working hours and meal consumption. The occurrence of pollution episodes in POA could be a consequence





of temperature-related biomass and trash burning (colder periods of day/year), agricultural burning (related to the Rabi and Kharif crop harvesting), fireworks, and bonfires (festivals, e.g., Lohri). POA is the largest contributor to these episodes at our site (Fig. S17).

It is also important to point out the differences in the HOA and BBOA factors identified in Spring 2018 (Figs. 4, S2f–g). Both HOA and BBOA exhibit episodes in the early morning hours of the day, with higher BBOA concentrations than HOA, and BBOA contribution to total organic mass as high as 38%. In the evening and at night, however, HOA is the larger of the two, contributing as much as 42% to total organic mass. Both components display strong diurnal behavior.

## 3.2 Oxidized organic aerosol (OOA)

### 3.2.1 Organic-only PMF mass spectra and diurnal patterns

Here, we discuss the mass spectral profiles of oxidized organic aerosol from the seasonal organic-only PMF analysis as well as the combined organic-inorganic PMF analysis. As shown in Fig. 1b, the observed OOA MS profile averaged across seasons is in line with the reference OOA profile. In every season, mass spectra of the OOA factors in organic-only PMF correlate strongly with the reference OOA factor (R>0.95) (Fig. S1a–f). The mass spectral profiles for each factor are provided in Fig. S5a–f. A large fractional contribution at $m/z$ 44 is a signature of the OOA factor. The fragment $CO_2^+$ dominates contributions at $m/z$ 44 compared to other fragments and is produced by the thermal decarboxylation and ionization-induced fragmentation of carboxylic acids (Ng et al., 2010). The large fractional contribution at $m/z$ 44 is beyond +1 standard deviation from the mean of the reference spectrum for most seasons (Fig. S5a-f). This higher contribution at $m/z$ 44 could be a result of rapid photochemical processing of fresh emissions, and/or regional transport of aerosol to the receptor site. The vicinity of the site to local sources suggests that rapid photochemical processing may be more important in causing the high $m/z$ 44.

Figure 5 shows the diurnal cycle of OOA in different seasons. Mean and median OOA concentrations, similar in magnitude at most times of the day in every season, increase during the morning, dip midday, and then increase again at nighttime. At nighttime, OOA concentrations remain stable for several hours into the next day and then increase in the early morning. OOA concentrations increase in the morning hours (~0500–0900 hours) despite increasing atmospheric mixing, pointing to photochemical formation related to primary emissions from traffic and other sources. The midday dip is likely associated with increasing atmospheric mixing and perhaps a reduction in daytime primary emissions. In contrast, nocturnal OOA concentrations are considerably less variable up to about 0400–0500 hours, perhaps representing a lower production rate in the absence of photochemistry. Overall, the timing of local primary emissions, meteorology, and boundary layer dynamics are likely responsible for differing behavior at different times of the day.

The diurnal variations of the OOA concentrations are stronger in winter (peak concentration ~1.9–2.3× midday concentration) than in the summer and monsoon (peak concentration ~1.6–1.7× midday concentration). Also, there is a strong seasonality of the factor with peak OOA concentrations in winter nearly 1.7× those in spring and ~2.9–3.6× those during the summer. Nevertheless, diurnal variations in OOA are weaker than their POA counterparts across seasons. Apart from a weaker diurnal variation, the daytime peaks in OOA are larger than nighttime peaks, and minima in OOA concentrations vary substantially





across seasons. As Fig. 5 shows, absolute OOA concentrations have large differences between winters and other seasons. In winters, the mean and median OOA concentrations never dip below 30 μg m$^{-3}$, whereas in other seasons they do not increase above 30 μg m$^{-3}$ except for early morning and at night in the spring and monsoon. As for interannual variability, winter and spring 2018 exhibit variations similar to the respective seasons in 2017.

Except for a brief period in the summer morning, we observe small differences between median and mean OOA concentrations at different hours of the day across seasons (Fig. 5). Similar conclusions can be drawn from the magnitude of SD compared to AM and GSD compared to GM (Table 3). Pollution episodes occur in primary emissions (Sect. 3.1); however, as shown here, this behavior does not translate to OOA.

### 3.2.2    Combined organic-inorganic PMF mass spectra and diurnal variation

Incorporating mass spectral data for inorganics in PMF allows further separation of OOA into AS-OOA, AN-OOA, and AC-OOA. In this section, we show that chloride, nitrate, and sulfate measured at the site are largely inorganic—as indicated by combined organic-inorganic PMF analysis which groups these species into factors with ammonium. Whereas, the association of organics with inorganics in PMF is a result likely arising from similar volatility. In previous work, OOA associated with ammonium nitrate usually has a mass spectrum showing higher fractional contribution at $m/z$ 43 (mostly $C_2H_3O^+$), than $m/z$

44 ($CO_2^+$), where $m/z$ 43 is believed to be associated with non-acid oxygenates (Ng et al., 2010). Additionally, these organic aerosols generally have time series reflecting semi-volatile behavior and are labeled semi-volatile oxidized organic aerosol (SVOOA) (Zhang et al., 2011). In this study, while AN-OOA exhibits semi-volatile behavior (Fig. 6a), the distribution of AN-OOA profiles and reference SVOOA are different (R<0.8, all seasons, Fig. S7a–f). Instead, the relative mass spectral contributions are in line with the reference OOA and the LVOOA profiles (R>0.95) (Figs. S7a–f, Figs. S18–S19a–f). The

strong mass spectral correlation of AN-OOA with reference LVOOA hints at rapid photochemical aging of aerosols in Delhi so that freshly formed oxidized aerosols have a high oxidation state. The mass spectrum of AS-OOA also compares well to the reference OOA and LVOOA profiles (R>0.95, all seasons) (Figs. S5a–f, S20-21a–f). AN-OOA has higher mean mass spectral contribution at $m/z$ 43 compared to AS-OOA; however, the mass spectral contributions at $m/z$ 44 are higher in AS-OOA. Additionally, at most high $m/z$s between 70–120, AN-OOA displays higher contributions than AS-OOA, which is likely

a result of less processing (Fig. S22a–b) (Ng et al., 2010). AS-OOA profiles have a higher contribution at other prominent lower $m/z$s, such as 17, 24, 41, and 55, as well.

AN-OOA and AS-OOA show different time series patterns which change both diurnally and seasonally. Figure 6a–b show the diurnal variations of AN-OOA and AS-OOA factor time series obtained from combined organic-inorganic PMF analysis. Based on the peak-to-minimum ratio in diurnal variations, AN-OOA shows stronger variations than OOA and AS-OOA (Fig.

6a–b). Like OOA, the smaller nighttime increase in AN-OOA compared to the daytime peak is likely due to the minimal photochemical formation at night. Also, like OOA, concentrations of AN-OOA are higher in winter than in other seasons, with concentrations never dipping below 12.5 μg m$^{-3}$ in winters. Among other seasons, AN-OOA concentrations exceed this value only in spring mornings. Within 2017, the diurnal patterns and the absolute AN-OOA concentrations change dramatically. In winters, summer, and monsoon, peak AN-OOA concentrations are ~2–5× the minima; in springs, this ratio increases to ~6–8×



. In terms of absolute variations in meteorological variables such as temperature, relative humidity (RH), wind speed, and VC, spring shows the most variability (Fig. S14). These variations might be causing the large peak-to-minimum ratio and the sudden drop in AN-OOA concentrations from winter to spring. The two winters and springs show very similar AN-OOA concentration variations. The mean-median difference, GSD versus GM, and SD versus AM point to episodic behavior in all seasons except

winters, possibly due to the semi-volatile behavior of AN-OOA leading to its evaporation at higher temperatures each day (Table 4).

On the other hand, AS-OOA exhibits flatter profiles, especially in the warmer months, with no prominent troughs and a single morning crest only in the colder seasons (winters and springs) (Fig. 6b). Concentrations in winters and springs peak between 0900–1000 hours, at ~2.0–2.5× the minima in both winters and ~1.5–1.8× in springs. Warmer seasons have similar

peak-to-minima ratios of ~1.5–1.7× in summer and ~1.7–2.0× in monsoon but exhibit flatter profiles in terms of absolute AS-OOA concentrations with mean and median always less than 8 μg m$^{-3}$. Additionally, high RH>60% occurs in winters and springs (Fig. S14), and 95$^{th}$ percentile episodes for AS-OOA occur predominantly at high RH (Fig. S23). Similar behavior has been observed for the ammonium sulfate aerosols previously measured in Delhi and elsewhere, particularly in the presence of high organic concentrations, and has been attributed to aqueous phase chemistry (Hu et al., 2011; Wang et al., 2016; Jaiprakash

et al., 2017; Sarangi et al., 2018; Wang et al., 2018). The flat and lower AS-OOA concentration levels during the rest of the day are likely an indicator of the contribution of regional sources to AS-OOA. AS-OOA diurnal profiles capture important processes such as nighttime secondary formation that is perhaps related to biomass burning emissions (winter 2017), morning peaks hinting at the possibility of rapid secondary formation from traffic and biomass burning emissions (winters and springs of 2017 and 2018), and prominent nighttime episodes (winter 2018). Interannual behavior of AS-OOA shows a decrease in

winter median throughout the day in 2018 compared to 2017 (lower by as much as 9 μg m$^{-3}$), and an increase in spring 2018 median AS-OOA concentrations compared to the median in spring 2017 (higher by as much as ~4 μg m$^{-3}$). The mean–median difference also suggests some midday episodes in spring 2017; these episodes are not observed in spring 2018. Overall, AS-OOA is less volatile than AN-OOA and shows weaker episodic behavior based on AM versus GM, SD versus AM, and GSD versus GM (Table 4).

Broadly speaking, springs have a larger AS-OOA to AN-OOA ratio relative to the winters and summer. This behavior is likely associated with the transitional meteorology of the spring period, where relatively low temperatures and weak atmospheric mixing coincide with higher photochemistry and relative humidity. Also, GM and median AN-OOA levels in Delhi are higher than AS-OOA only in winters, pointing to the strong influence of temperature on PM formation (the evaporation of semi-volatile compounds in warmer periods), and the importance of less photochemistry in winters. To summarize, despite overall

consistency across seasons for OOA, its components AN-OOA and AS-OOA show very different behavior, which changes both diurnally and seasonally.

Akin to the associations with ammonium nitrate and ammonium sulfate, some organic components also associate with ammonium chloride (Fig. S7a–f). These organics (AC-OOA) account for 17–23% mass of this factor (ACOOA). The median fraction of organics associated with this factor does not exceed 4%. Unlike AS-OOA and AN-OOA, the mass spectra of these

organics are not associated with a specific category of reference OA type. Instead, they resemble oxidized biomass burning



aerosol (higher fraction of organics at $m/z$ 44 than reference BBOA profile) (Figs. S7a–f, S24a–h). These organics may be associated with the AC factor due to their semi-volatile behavior. Indeed, chloride has been used as a tracer for semi-volatile OOA (Zhang et al., 2011). Additionally, chloride has been used as a tracer for biomass burning, specifically agricultural burning and more recently, has also been detected in plastic burning (Li et al., 2014a, b; Kumar et al., 2015; Fourtziou et al., 2017).

However, as Fig. S6a–b and Fig. S6e–f show, for the four seasons (Winters 2017 and 2018, Springs 2017 and 2018), when significant chloride mass is detected in the particulate phase, the time series of the biomass burning tracers, aethalometer-derived biomass burning component of black carbon, $BC_{BB}$, and the difference between UVPM and BC, $\Delta C$, do not correlate with this AC factor. Additionally, this factor shows strong directionality, with 87% of the 95[th] percentile episodes occurring when winds are blowing from the North-Northwest (N-NW) direction compared to about 70% for other factors (Fig. S25e).

Since Delhi is in an especially ammonia-rich environment (Warner et al., 2017; Van Damme et al., 2018), this observation suggests that ammonium chloride is forming from the interaction of upwind chloride sources and fertilizer emissions. One logical source of chloride emissions would be the industrial use of hydrochloric acid (Jaiprakash et al., 2017).

Considering that the combined organic-inorganic PMF results cluster nitrate, chloride, and sulfate with ammonium, our findings suggest that nitrates, chlorides, and sulfates measured by the ACSM in New Delhi are dominated by inorganics,

particularly at higher concentrations. This finding is consistent with results from analysis based on tracer ratios. The $ON_{NO_x}$ method, developed by Farmer et al. (2010), estimates organic nitrate (ON) contribution in the $NO_3$ detected by online mass spectrometers such as the AMS based on the ratio of $NO_3$ fragments at $m/z$ 30 to 46. The higher the value relative to the pure ammonium nitrate calibration, the greater the importance of organonitrates. The $m/z$ 30 to 46 ratio increases from winter to monsoon and does not exceed the calibration ratio (3.6) in winters, pointing to the small fraction of organonitrates at high nitrate

concentrations. Similarly, Wang and Hildebrandt Ruiz (2017) suggest the use of the ratio of chloride $m/z$ 35 to $m/z$ 36 mass for estimating organochloride presence. We observe extremely stable values of this ratio between 0.05–0.3 across all seasons and hours of the day when chloride is in particle phase, pointing to the dominance of inorganic chlorides. Finally, Hu et al. (2017) suggest the use of $SO^+$ to $SO_3^+$ ratio to estimate the presence of organosulfur species. The range of seasonally representative diurnal median across seasons is 2.4–4.1, agreeing very well with the median values obtained with pure ammonium sulfate

calibrations (3.0). This result points to the dominance of inorganic sulfate over organosulfates across seasons. Although the diurnal patterns are stable across seasons, there is only a slight upward shift in the median ratio from winter 2017 to spring 2018, pointing to minimal changes in the importance of organosulfates across seasons.

To summarize, our findings indicate that AN-OOA shows stronger diurnal variability than AS-OOA and AS-OOA to AN-OOA ratio is high in spring likely due to transitional meteorology. Compared to other factors, AS-OOA concentrations show

a strong association with high RH, possibly a consequence of aqueous phase chemistry. While chloride has been used as a tracer for biomass burning, we do not see any correlation with the aethalometer-derived biomass burning tracers, and suspect chloride to be of industrial origin. Finally, based on PMF results and tracer ratios for organic nitrates, chlorides, and sulfates, these species are principally inorganic.



### 3.3 Primary versus secondary organics

In this section, we summarize the contributions of organics as either primary or secondary using the PMF results. We show that (i) primary emissions are more important in high PM pollution periods, (ii) secondary organic aerosols dominate average concentrations year-round, and (iii) a modified tracer-based organic component evaluation could provide real-time source apportionment of POA and SOA for Delhi.

For fractional POA contributors, the nighttime peak between 1800–2200 hours is larger than the daytime peak between 0500–0800 hours, likely due to reduced photochemistry (Fig. 7b). The minima in the variations occur between 1200–1500 hours in all seasons, likely due to a combination of reduced sources, increased ventilation, and higher temperatures (Fig. 7b). For winter 2017, although the fraction of POA is generally less than 50%, it nears or exceeds the halfway mark during the early morning traffic and sunrise hours (0600–0800 hours) and for most hours in the evening and at night (1700–0100 hours), with similar behavior persisting across seasons. These time windows also correspond to periods with the highest POA and highest total concentrations, pointing to the importance of local, primary emissions in the high pollution periods, within a day, within each season and across seasons (Fig. S17). Relative to colder seasons of winter and spring, warmer months exhibit similar midday POA fractions but diverging lower nighttime fractions. Spring 2018 POA fraction varies between 34–75%, much higher than the winters (20–68%), spring 2017 (27–62%), and summer 2017 and monsoon 2017 (23–53%). Fractional contributions of OOA to total NRPM$_1$ exhibit very similar timing of crests and troughs across seasons (Fig. 7a). However, OOA fractions for different seasons converge in the middle of the day and diverge in the early morning and at night, with nighttime fractions in colder months generally lower than the warmer months. This result likely reflects the generally greater influence of primary emissions at night and during colder months. The OOA fractions peak between 1400–1600 hours at ~66-80%. The profiles rapidly descend to minima between 25–51% between 1900–2000 hours, likely due to lowered photochemistry and source strength. As for interannual variability, winter 2018 has lower OOA fractions, particularly at nighttime, by as much as 14%. Further, springs of 2017 and 2018 show large differences in the early morning and at night with OOA in 2018 always contributing less (up to 26% less around midnight). This difference could, in part, be due to the separation of a BBOA factor in spring 2018 that allows deconvolution of the biomass burning component. Despite the changes, the diurnal patterns remain consistent interannually.

We have also compared full PMF results for organic mass with a modified tracer-based organic component evaluation (Ng et al., 2011a). The previously deployed tracer approach utilizes linear scaling of mass at specific *m/z*s to estimate the total mass of each of the three factors—HOA, BBOA, and OOA. Since full PMF mostly yields two factors—proxies for primary and secondary aerosols, the comparisons are conducted for primary organic aerosols (POA=HOA+BBOA in our analysis) and secondary organic aerosols (OOA) only. However, for our measurements, this results in the total mass estimated using the tracer-based approach substantially different from the actual organic mass (POA: slope~0.29, intercept~5.4, R~0.87; OOA: slope~0.41, intercept~17.3, R~0.55) (Fig. S26). Accordingly, a mass closure on the factors has been applied, which ultimately affects the PMF factor concentrations as well. For hourly averaged data, the tracer-based factors are strongly linearly correlated with results from the full PMF with slopes close to 1 (POA: slope~0.91, intercept~−3.7, R~0.98; OOA: slope~1.09,





intercept~3.9, R~0.96) (Fig. 8). Given that the tracer-based approach can be run almost instantly, this approach has the potential as an effective real-time source apportionment approach curated for Delhi.

## 3.4 Factors influencing organic aerosol concentrations and composition

### 3.4.1 Effect of meteorological variables on interannual variability

In this section, we test the association of the interannual increase in concentrations with meteorological variables. Apart from source effects, the changes can be a consequence of changing wind direction, wind speed, temperature, RH, and boundary layer height. Here, we compare the seasonally averaged diurnal patterns of these variables (Fig. S27a–d). RH averaged over 2017 winter is up to 10% higher compared to winter 2018 (Fig. S27a). However, RH primarily affects the ASOOA factor in high pollution episodes (Fig. S23). At the same time, the temperature between the two winters is within 2°C of each other (Fig.

S27b). In spring, RH is within 6%, and temperature within 4°C at all hours. Diurnal patterns of wind direction are similar in 2017 and 2018, except in the middle of the day when there is a significant shift of about 28° from N-NW to N-NE in winter (Fig. S27c). Between springs also, there is a slight shift of about 14° towards N-NE late in the evening and night between 2000–2400 hours and about 29° between 0500–0800 hours. Ventilation coefficients are consistent across the two years, except at nighttime—nighttime median VCs are lower by as much as 80% in winter and spring of 2018 (Fig. S27d). Thus, a lower VC

for hours of the day when HOA and BBOA are most important could be a key reason behind their higher concentrations. The increase could also be due to an enhancement of sources. Specifically, in winters, it could also be attributed to the extra period sampled—winter 2018 samples between December 21 and January 14, a period missed in winter 2017.

### 3.4.2 Effect of ventilation-related variables on factor concentrations

  The association of ventilation related variables with organic-inorganic combined PMF factors shows that (i) wind speed does

not necessarily provide ventilation, and (ii) boundary layer dynamics seem to regulate all components, but to different extents. For all factors, pollution increases with wind speed up to 3.75 ms$^{-1}$ (Fig. S28a–e). This observation is important and points to the importance of winds bringing an influx of pollutants in Delhi rather than providing "ventilation". Winds seem to enhance episodes of primary emissions—POA and AC—stronger than the oxidized components. This result might be attributable to incoming wind flow bringing in primary emissions from areas upstream of Delhi that are emission hotspots replete with sources,

which has been suggested recently (Jaiprakash et al., 2017). The other component important for "ventilation", the planetary boundary layer height (PBLH), reflects the disparate response of factors (Fig. S29a–e). Going from the 5[th] to 95[th] percentiles, percent of chloride episodes for PBLH<100 m increases from 39% to 85%. In comparison, POA episode fractional contribution increases from about 27% to 66%, whereas AN-OOA increases from about 28% to 41%, and AS-OOA increases from about 26% to 45%. However, temperature and ventilation coefficient are correlated (R~0.6), which implies that the association with

boundary layer height is likely convoluted with an association with temperature. Finally, the lowest ventilation coefficient up to 500 m$^2$s$^{-1}$ displays increasing clustering for higher percentiles of episodes as the selection is made from the 5[th] to the 95[th]





percentile (Fig. S30a–e). The increase is from 40% to 88% for AC, about 31% to 70% for POA and about 30% to 50% for AN-OOA and AS-OOA. Clearly, the modulation is stronger for primary components than secondary components.

## 4 Conclusions

This study provides long-term source apportionment results for a receptor site in New Delhi, the most polluted megacity in the world. For the first time, high time resolution data is available for understanding diurnal patterns and seasonal and interannual changes of submicron primary and secondary aerosols. Organic-only PMF analysis yields 2–3 factors—POA (HOA+BBOA) and OOA in every season and HOA and BBOA separating as factors only in spring 2018. However, mass spectral contribution at biomass burning tracers and, correlations with reference MS profiles suggest the influence of biomass burning, especially in colder months. Among factors occurring in every season, POA exhibits the strongest diurnal patterns, with the nighttime peak much larger than the daytime peak. The combined organic-inorganic PMF analysis allows separation of oxidized organic aerosols into components based on association with ammonium nitrate and ammonium sulfate. The two components display different diurnal patterns, with AN-OOA showing relatively stronger diurnal patterns and AS-OOA displaying flatter profiles with a sharp rise and descent in the middle of the day, likely associated with photochemical formation. AS-OOA shows a sharp increase in concentrations, especially for 95$^{th}$ percentile episodes at high RH, in line with recent observations of aqueous phase chemistry. AN-OOA mass spectral profiles are very similar to reference LVOOA profiles, despite vicinity to local sources, pointing to the rapid processing of aerosols in Delhi. Chloride, nitrate, and sulfate are mostly inorganic. Analysis using the volatility basis set suggests that differences in temperature and, therefore, equilibrium partitioning, can explain differences in observed winter and summer concentrations. While temperature and ventilation coefficient play a crucial role in determining the organic presence in gas or aerosol phase, the N-NW direction is associated with 70% of the 95$^{th}$ percentile episodes for all factors except AC-OOA, for which it further increases to 87%. Increasing wind speed up to 3.75 ms$^{-1}$ is associated with an enhancement of concentrations, likely due to the influx of emissions from upstream. Observations indicate that regional OA and ammonium chloride contribute substantially to air pollution in Delhi. Further, primary aerosols such as POA and AC increase whereas secondary aerosols such as AN-OOA and AS-OOA stabilize in the high pollution periods. Thus, primary species are more important contributors to the high pollution episodes in Delhi despite secondary organics dominating year-round fractional contributions to total organics.

The current Indian legislative and legal framework tasks the responsibility of managing air pollution in Delhi to the Environmental Pollution (Prevention & Control) Authority. The Graded Action Response Plan (GRAP) currently in place accounts for a multitude of local and regional sources contributing to air pollution in Delhi (Ministry of Environment, Forest & Climate Change, Government of India, 2018). However, the lack of chemically speciated emission inventories limits the feasibility of implementation of such broad programs restricting economic activity. The results presented here capture the complexity of aerosols in Delhi by speciating them into different chemical categories and identifying associations of those components with each other and meteorological variables and evaluating their behavior at different times of the day and the year. Because the data presented is highly time-resolved, it can provide critical insight into the diverse sources that contribute to pollution loadings,





and similar techniques could be used to measure the efficacy of air pollution regulatory actions. Results from this work address several pressing requirements for air quality management in Delhi including a real-time source apportionment approach for organics, evidence of extremely high chloride concentrations associated with industrial sources, and providing evidence differentiating high pollution episodic events from seasonal averages. Future work could utilize these speciated measurements to
delve into the identification of potential source locations using back-trajectory analysis.

*Data availability.*   Underlying research data will be available in the Texas Data Repository (https://data.tdl.org/) upon publication of the final manuscript.

## Appendix A:  Abbreviations

AC: Ammonium Chloride; AC-OOA: Oxidized Organic Aerosol associated with Ammonium Chloride; ACSM: Aerosol Chem-
ical Speciation Monitor; AMS: Aerosol Mass Spectrometer; AN: Ammonium Nitrate; AN-OOA: Oxidized Organic Aerosol associated with Ammonium Nitrate; AS: Ammonium Sulfate; AS-OOA: Oxidized Organic Aerosol associated with Ammonium Sulfate; BBOA: Biomass Burning Organic Aerosol; BC: Black Carbon; CO: Carbon Monoxide; CPCB: Central Pollution Control Board; DAS: Delhi Aerosol Supersite; HOA: Hydrocarbon-like Organic Aerosol; HYSPLIT: NOAA Hybrid Single Particle Lagrangian Integrated Trajectory Model; IIT: Indian Institute of Technology; MS: Mass Spectra; NCR: National Cap-
ital Region; N-NW: North-Northwest; $NRPM_1$: Non-refractory Submicron Particulate Matter; ON: Organic Nitrate; $ON_{NO_x}$: Method to estimate organonitrate fraction using fragment ratio of $NO_3$ 30 to 46; OOA: Oxidized Organic Aerosol; PBLH: Planetary Boundary Layer Height; PET: PMF Evaluation Tool; $PM_1$: Submicron Particulate Matter; $PM_{10}$: Particulate matter smaller than 10 μm in diameter; PMF: Positive Matrix Factorization; POA: Mixed HOA BBOA; RH: Relative Humidity; SOA: Secondary Organic Aerosol; UVPM: Ultra-Violet absorbing Particulate Matter; VBS: Volatility Basis Set; VC: Ventilation
Coefficient; WHO: World Health Organization;

*Author contributions.*   LHR, JSA, GH, SB, and SG designed the study. SG, SB, PS, and ZA carried out the data collection. SB, KP, DSW, and SG carried out the data processing and analysis. SB, SG, KP, JSA, and LHR assisted with the interpretation of results. All co-authors contributed to writing and reviewing the manuscript.

*Competing interests.*   The authors declare that they have no conflict of interest.

*Acknowledgements.*   JSA was supported by the Climate Works Foundation. We are thankful to the Indian Institute of Technology Delhi (IITD) for institutional support. We are grateful to all student and staff members of the Aerosol Research Characterization laboratory (especially



Nisar Ali Baig and Mohammad Yawar) and the Environmental Engineering laboratory (especially Sanjay Gupta) at IITD for their constant support. We are thankful to Philip Croteau (Aerodyne Research) for always providing timely technical support for the ACSM.



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





**Table 1.** Seasonal summary of PM$_1$ species—the arithmetic mean (AM) for hourly mass concentrations (μg m$^{-3}$). Adapted from Gani et al. (2018).

|          | Winter | Spring | Summer | Monsoon |
|----------|--------|--------|--------|---------|
| Org      | 112    | 61     | 35     | 23      |
| NH$_4$   | 20     | 10     | 5.2    | 4.6     |
| Chl      | 23     | 9.5    | 1.5    | 0.4     |
| NO$_3$   | 24     | 9      | 3.8    | 3.6     |
| SO$_4$   | 16     | 10     | 10     | 10      |
| BC       | 15     | 11     | 9      | 11      |
| NR-PM$_1$| 195    | 100    | 55     | 41      |



**Table 2.** Periods for seasonal PMF analysis.

| Season | Dates |
| --- | --- |
| Winter 2017 | Dec 1–Feb 14 |
| Spring 2017 | Feb 15–Mar 31 |
| Summer 2017 | Apr 1–Jun 30 |
| Monsoon 2017 | Jul 1–Sep 15 |
| Winter 2018 | Dec 1–Feb 14 |
| Spring 2018 | Feb 15–Mar 31 |



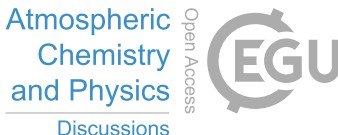

**Table 3.** Organic-only PMF factor concentrations (µg m$^{-3}$) split into POA and OOA.

|  | Arithmetic Mean (SD) | | Median | | Geometric Mean (GSD) | |
|---|---|---|---|---|---|---|
|  | POA | OOA | POA | OOA | POA | OOA |
| Winter 2017 | 52 (48) | 56 (21) | 35 | 56 | 36 (2.5) | 51 (1.5) |
| Spring 2017 | 30 (30) | 30 (16) | 20 | 27 | 20 (2.5) | 26 (1.8) |
| Summer 2017 | 15 (16) | 19 (15) | 8 | 16 | 9 (2.5) | 15 (2.1) |
| Monsoon 2017 | 9 (8) | 15 (9) | 6 | 13 | 7 (2.3) | 12 (1.9) |
| Winter 2018 | 61 (56) | 56 (24) | 41 | 54 | 39 (2.7) | 51 (1.6) |
| Spring 2018[a] | 20 (19); 21 (18) | 24 (12) | 12; 23 | 17 (4, 67) | 13 (2.6); 15 (2.6) | 21 (1.7) |

[a]For spring 2018, we were able to separate POA further into HOA and BBOA factors.





**Table 4.** Organic component of the organic-inorganic combined PMF factor concentrations (μg m$^{-3}$) split into POA and OOA. For OOA, we were able to separate OOA into AN-OOA, AS-OOA, and AC-OOA.

|  | Arithmetic Mean (SD) | | Median | | Geometric Mean (GSD) | |
|---|---|---|---|---|---|---|
|  | POA | OOA | POA | OOA | POA | OOA |
| Winter 2017 | 60 (57) | 23 (12); 19 (10); 6 (7) | 38 | 22; 18; 4 | 38 (3.3) | 19 (2.5); 15 (2.2); 3 (5.1) |
| Spring 2017 | 31(33) | 9 (9); 14 (6); 4 (6) | 20 | 6; 13 ; 2 | 19 (2.9) | 5 (3.5); 12 (1.7); 1 (5.6) |
| Summer 2017[a] | 16 (18) | 8 (8); 6 (4) | 9 | 5; 5 | 10 (2.7) | 4 (4.1); 5 (2.2) |
| Monsoon 2017[a] | 11 (9) | 2 (2); 4 (3) | 8 | 1; 3 | 8 (2.5) | 1 (3.4); 3 (2.0) |
| Winter 2018 | 68(62) | 22 (12); 15 (11); 12 (13) | 42 | 20; 12; 6 | 44 (2.7) | 19 (1.9); 12 (2.4); 6 (3.4) |
| Spring 2018 | 38(36) | 7 (7); 16 (7); 5 (8) | 24 | 4; 15; 1 | 24 (2.9) | 4 (3.4); 14 (2.1); 1 (6.9) |

[a]For summer and monsoon, OOA was separated only into AN-OOA and AS-OOA as there was no AC-OOA factor.

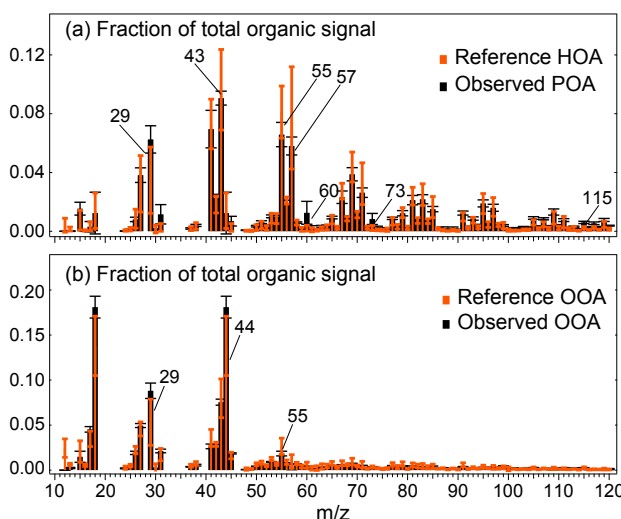

**Figure 1.** a) The average mass spectrum of organic-only PMF primary organic aerosol (POA) factor shows influence of both hydrocarbon-like organic aerosols (HOA) and biomass burning organic aerosols (BBOA). b) The average mass spectrum of organic-only PMF oxidized organic aerosol (OOA) factor, which is similar to the reference OOA factor. The whiskers in the graphs represent ±1 standard deviation from the mean spectra.





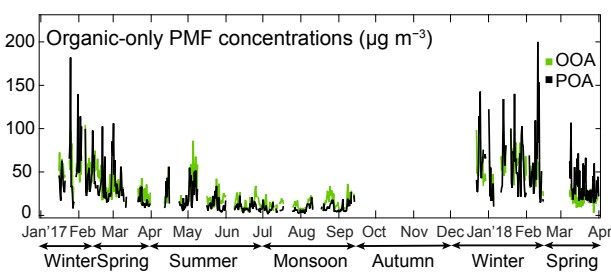

**Figure 2.** The 12 h averaged concentration time series (TS) of organic-only PMF oxidized organic aerosol (OOA) factor and primary organic aerosol (POA) factor ($\mu$g m$^{-3}$). Factor concentrations show strong seasonal and diurnal variations.

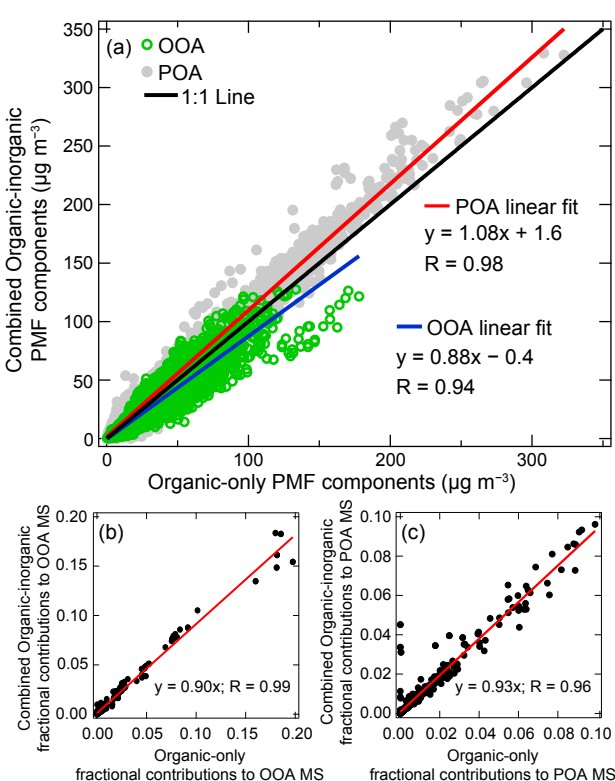

**Figure 3.** a) Time series correlation of POA and OOA from organic-inorganic PMF and organic-only PMF analysis (µg m$^{-3}$) b) and c) Mass spectral correlations of OOA and POA from organic-inorganic PMF and organic-only PMF analysis. Strong correlations suggest an agreement between the analysis.





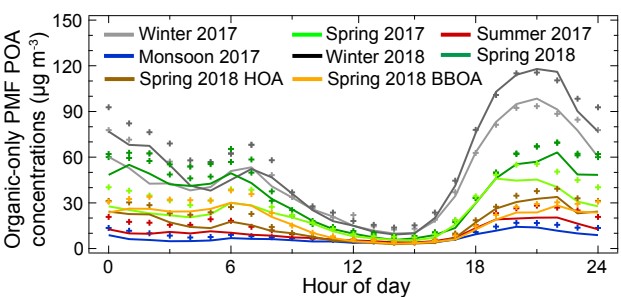

**Figure 4.** Seasonally representative diurnal mean (+) and median concentrations (lines) of POA from organic-only PMF analysis. This figure shows that POA exhibits extreme diurnal variations and displays two peaks corresponding to the early morning and late evening traffic hours.



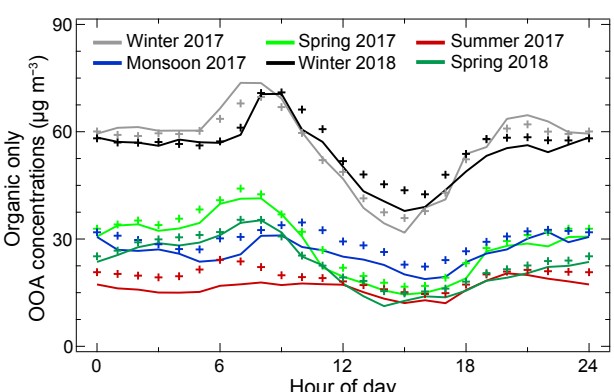

**Figure 5.** Seasonally representative diurnal mean (+) and median concentrations (lines) of OOA from organic-only PMF analysis. This figure shows the higher concentrations of OOA in winters compared to other months and the relatively stronger diurnal variations in winters and springs compared to other seasons.

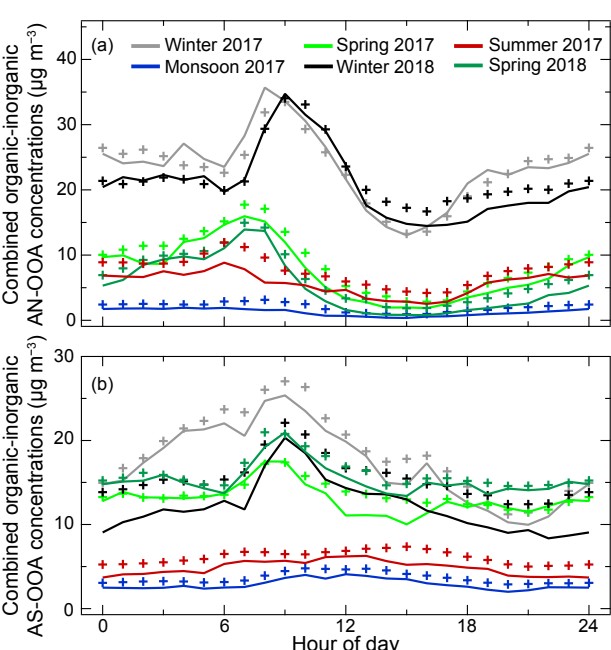

**Figure 6.** Seasonally representative diurnal mean (+) and median (lines) concentrations of (a) AN-OOA (b) AS-OOA from organic-inorganic combined PMF analysis. Compared to AN-OOA, AS-OOA concentrations remain relatively stable throughout the day, particularly in warmer months.



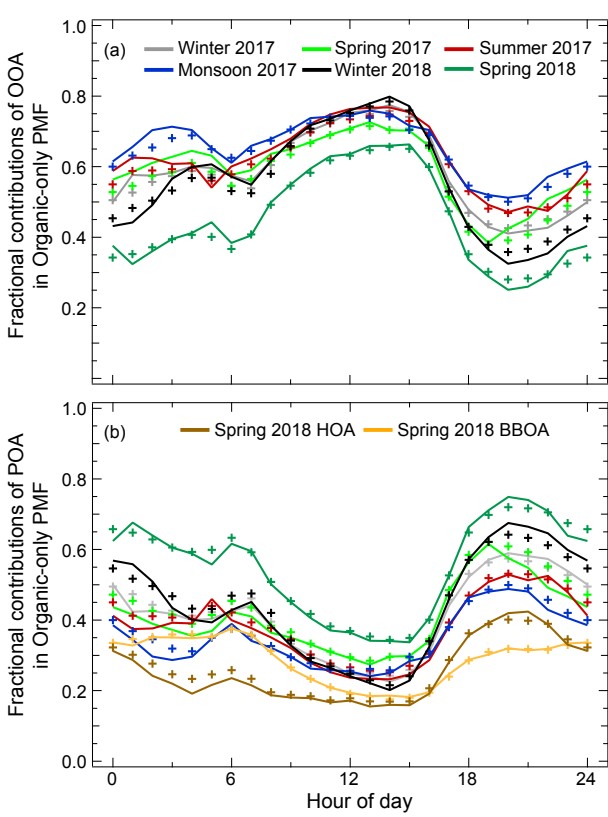

**Figure 7.** a) Diurnal mean (+) and median (lines) fraction of OOA from organic-only PMF analysis in different seasons. Fractional contributions vary between 32–80%, except spring 2018. This is likely due to the separation of BBOA mixed into the oxidized organics in other seasons. b) Diurnal mean and median composition of POA from organic-only PMF analysis. Spring 2018 has a relatively higher POA fraction, due to the unmixing of a separate BBOA profile.



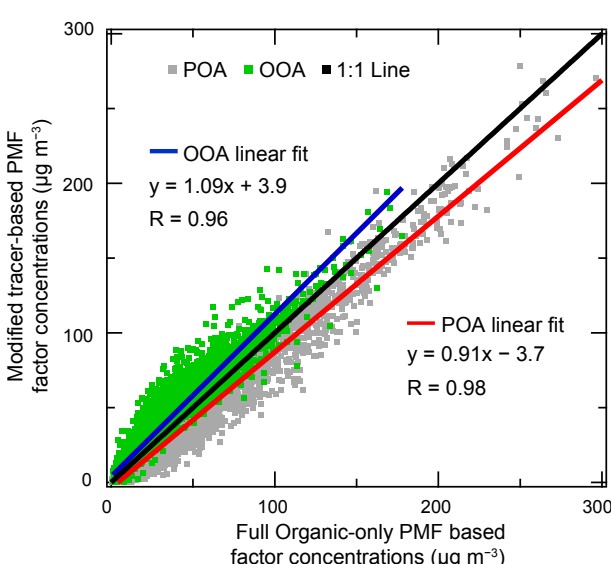

**Figure 8.** This figure shows the comparison of the modified tracer-based OOA and POA concentrations with those obtained from organic-only PMF. The strong linear correlation and slope close to 1 point to the usefulness of the tracer approach for real-time source apportionment.