# Peer review of "Sources and atmospheric dynamics of organic aerosol in New Delhi, India: Insights from receptor modeling"

_Atmospheric Chemistry and Physics, 2019_

## Referee Comment (RC1) · Anonymous Referee #2 · 22 Jul 2019

General comment The paper is focused on characterisation of organic aerosol in New Delhi (India) using an ACSM and some additional measurements. The approaches used with PMF receptor model allow to get some information about the sources and the trends in primary and secondary organics. The topic is interesting, up-to-date and suitable for the Journal. In general, it is not easy to read this paper because most of the figures and results are reported in the supplementary material (73 pages) that seems to be richer than the main paper. Authors should think about bringing some of the main results in the paper removing them from the supplementary material. His at least for the results that authors say to be important like the relationships with PBL height or the influence of the ventilation coefficient. Further, there are some aspects that are not

clear (see my specific comments) that should be addressed in a revision step.

Specific comments

Abstract. Thermodynamic modelling. What kind of modelling and for what?

Abstract (and also in the main text). Authors speak of inter-annual variability, however, having only 15 months of measurements, the possibility to see an inter-annual trend is optimistic to say the least. I would suggest to change this aspect.

Page 2 (line 8). Molecular markers such as...?

Page 2 (lines 21-32). I agree that the high temporal resolution could furnish additional information compared to receptor models applied to 24h samples. However, in the case of using ACSM only some chemical species are available and there are limitations due to the missing information on metals and other compounds (for example levoglucosan and similar). This aspect should be clearly mentioned and a reference to the recent work of Belis et al (Atmospheric Environment 123 (2015) 240e250) regarding PMF and receptor models performances should be added.

Section 2 (methods). ACSM is working on PM1 instead, other measurements have been done on PM2.5. Why not on the same size fraction? The differences should be explained. In addition, BC, and UVPM are not shown at all in the main paper but only in the supplementary material is this meaning that these species are not so important in the framework of the results?

Page 3 (section 2.1). Fifteen months divided into six seasons, why not using calendar seasons, I mean one year divided in four seasons?

Section 2.2. How it is used CO? In the PMF? Again, no trace of CO is reported in the main text.

Page 5 (lines 26-28). It is reported that OOA correlates strongly with both sulphate and nitrate. However, in several sites nitrate and sulphate have different seasonal

trends with sulphate larger in the warm seasons and nitrate larger in the cold seasons because of its thermal instability. This is true for both ammonium nitrate and sodium nitrate coming from aged marine salt. If I have well understood there is a correlation between SOA with both nitrate and sulphate suggesting that at this they have the same trend. Is this true? In case it will be useful to discuss this aspect mentioning explicitly the similarity in these trends that is not often observed at other sites.

Page 5 (line 30-31). Actually, looking at Figure 2 POA and OOA seems quite comparable. Are the difference mentioned statistically significant?

Page 7 (lines 9-10). This sentence is not correct because at night there is not a decreasing PBL, rather at sunset a new shallow boundary-layer is established generally thermally stable that could trap pollutants and will evolve at sunrise the day after mixing air from ground level with air masses above, see for example Meteorologische Zeitschrift, Vol. 21, No. 4, 385-398 (August 2012). I believe that authors should explain better this part.

Page 12 (lines 14-15). These percentages are really so different statistically?

Page 12 (lines 18-19). The larger primary emissions during cold months is not reflected in the percentages above because the maximum appears to be during spring.

Figures 3b and 3c. The intercepts are missing, are they negligible?

Figures 4, 5, 6, and 7. Please report in the caption what is the difference between continuous lines and marks.

---

## Referee Comment (RC2) · Anonymous Referee #1 · 26 Sep 2019

The authors present a very valuable new dataset focusing on the organic fraction of submicron aerosol composition in Delhi, India from January 2017 to March 2018. Importantly, the dataset covers two winter seasons, which are the periods of most intense pollution in Delhi. To my knowledge this dataset is the first of its kind and provides significant insight into the nature and possible sources of PM2.5 in Delhi. I believe it should be published after a few minor points are addressed

- In some places including the abstract the authors have referred to this study as a "source apportionment." While it is clear that this data and analysis have provided very important insight into possible sources/formation pathways for OA in Delhi, "source

apportionment" is a stronger term. I think the authors would agree that they have not identified the fraction of OA in Delhi originating from various specific anthropogenic and biogenic sources.

- A lot of real estate in the paper is devoted to figures which illustrate the PMF analysis but do not convey take-home messages for the reader. Consider moving those to SI.

- The results clearly show elevated organic aerosol concentrations in winter. However, overall PM concentrations are also higher in winter. A figure (or figure panel) showing the seasonal variation in the OA fraction of PM mass would be very helpful

---

## Author Comment (AC1) · 7 Nov 2019

We thank the reviewers for their comments. All comments are addressed below. Reviewers' comments are included in italics, our responses are included in cyan, and updated manuscript text is included in red.

*Reviewer 1*

*The authors present a very valuable new dataset focusing on the organic fraction of submicron aerosol composition in Delhi, India from January 2017 to March 2018. Importantly, the dataset covers two winter seasons, which are the periods of most intense pollution in Delhi. To my knowledge this dataset is the first of its kind and provides significant insight into the nature and possible sources of PM2.5 in Delhi. I believe it should be published after a few minor points are addressed. In some places including the abstract the authors have referred to this study as a "source apportionment." While it is clear that this data and analysis have provided very important insight into possible sources/formation pathways for OA in Delhi, apportionment is a stronger term. I think the authors would agree that they have not identified the fraction of OA in Delhi originating from various specific anthropogenic and biogenic sources.*

Response: We agree with the reviewer that the analysis does not conduct source apportionment with regards to the differences between anthropogenic and biogenic sources. However, our analysis allows us to separate organic aerosol (OA) of primary versus secondary origin, and to separate primary traffic-related OA from biomass burning OA in Spring 2018. The limited number of primary sources identified is likely due to the unit mass resolution of the ACSM, as has been observed previously (Aiken et al., 2009; Elser et al., 2016; Al-Naiema et al., 2018).

Therefore, we suggest continuing with the use of the term "source apportionment" but have clarified the terminology in section 3, where we have added the following text:

"Thus, the source apportionment conducted was limited to separating primary and secondary OA in all seasons (and further, primary into BBOA and HOA in Spring 2018). Further separation of the factors may be possible with high-resolution data combined with application of factor constraints (Aiken et al., 2009; Elser et al., 2016; Al-Naiema et al., 2018) and source-specific measurements such as metals/metal ions for combustion or biomass burning emissions (Jaiprakash et al, 2017), or α-methylglyceric acid for biogenic secondary OA (Srivastava et al, 2019)."

*- A lot of real estate in the paper is devoted to figures which illustrate the PMF analysis but do not convey take-home messages for the reader. Consider moving those to SI.*

Response: Based on the reviewer's comment, we have edited the manuscript to emphasize key take-home messages of each figure. We also agree with the reviewer that "a lot of real estate in the paper is devoted to figures which illustrate the PMF analysis", which is the focus of this paper. We have moved Figs. 3 and 8 into the SI. The slopes and correlation coefficients shown in the figures have been inserted into the main text. The following are updated portions in the text (Section 3):

"It is therefore not surprising, as shown in Fig. S8a–c, that the behavior of POA and OOA in combined organic-inorganic PMF is very similar to that of organic-only POA and OOA respectively (POA: Slope~1.08, Intercept~1.6, R~0.98, OOA: Slope~0.88, Intercept~-0.4, R~0.94)."

The other figures are essential for supporting results presented in the paper. Figure 1 introduces the broad similarity of PMF factor analysis with that expected based on analysis conducted elsewhere (Ng et al., 2011, Zhang et al., 2011). Using time series patterns, Fig. 2 summarizes seasonal variations in primary and secondary PMF components using the large amount of data that was collected as a part of the campaign. Additionally, given that this is the first long-term PMF analysis published for organic MS in Delhi, we believe that developing an understanding of diurnal patterns of different aerosol types is very important. Accordingly, Figs. 4-7 elucidate diurnal patterns associated with PMF factors that reflect insights into possible sources and formation pathways that would otherwise be difficult to convey.

*- The results clearly show elevated organic aerosol concentrations in winter. However, overall PM concentrations are also higher in winter. A figure (or figure panel) showing the seasonal variation in the OA fraction of PM mass would be very helpful.*

Response: A figure detailing the fraction of species detected as "Org" or organics in the ACSM has already been published (Fig. 3, Gani et al (2019)).

We have updated the text to reference that figure:

"Over the campaign, organics accounted for 53% of the submicron mass, followed by inorganics (36%, of which sulfate, nitrate, ammonium, and chloride contributed 13%, 8%, 9%, and 6% respectively) and black carbon (BC) (10%) (Gani et al., 2019, Fig. 3)."

References:

Aiken, A. C., Salcedo, D., Cubison, M. J., Huffman, J. A., DeCarlo, P. F., Ulbrich, I. M., Docherty, K. S., Sueper, D., Kimmel, J. R., Worsnop, D. R., Trimborn, A., Northway, M., Stone, E. A., Schauer, J. J., Volkamer, R. M., Fortner, E., de Foy, B., Wang, J., Laskin, A., Shutthanandan, V., Zheng, J., Zhang, R., Gaffney, J., Marley, N. A., Paredes-Miranda, G., Arnott, W. P., Molina, L. T., Sosa, G., and Jimenez, J. L.: Mexico City aerosol analysis during MILAGRO using high resolution aerosol mass spectrometry at the urban supersite (T0) — Part 1: Fine particle composition and organic source apportionment, Atmospheric Chemistry and Physics, 9, 6633–6653, https://doi.org/10.5194/acp-9-6633-2009, 2009.

Al-Naiema, I. M., Hettiyadura, A. P. S., Wallace, H. W., Sanchez, N. P., Madler, C. J., Cevik, B. K., Bui, A. A. T., Kettler, J., Griffin, R. J., and Stone, E. A.: Source apportionment of fine particulate matter in Houston, Texas: insights to secondary organic aerosols, Atmospheric Chemistry and Physics, 18, 15 601–15 622, https://doi.org/10.5194/acp-18-15601-2018, 2018.

Elser, M., Huang, R.-J., Wolf, R., Slowik, J. G., Wang, Q., Canonaco, F., Li, G., Bozzetti, C., Daellenbach, K. R., Huang, Y., Zhang, R., Li, Z., Cao, J., Baltensperger, U., El-Haddad, I., and Prévôt, A. S. H.: New insights into PM2:5 chemical composition and sources in two major cities in China during extreme haze events using aerosol mass spectrometry, Atmospheric Chemistry and Physics, 16, 3207–3225, https://doi.org/10.5194/acp-16-3207-2016, 2016.

Gani, S., Bhandari, S., Seraj, S., Wang, D. S., Patel, K., Soni, P., Arub, Z., Habib, G., Hildebrandt Ruiz, L., and Apte, J. S.: Submicron aerosol composition in the world's most polluted megacity: the Delhi Aerosol Supersite study, Atmospheric Chemistry and Physics, 19, 6843–6859, https://doi.org/10.5194/acp-19-6843-2019, 2019.

Jaiprakash, Singhai, A., Habib, G., Raman, R. S., and Gupta, T.: Chemical characterization of PM1 aerosol in Delhi and source apportionment using positive matrix factorization, Environmental Science and Pollution Research, 24, 445–462, https://doi.org/10.1007/s11356-016-7708-8, 2017.

Ng, N. L., Canagaratna, M. R., Jimenez, J. L., Zhang, Q., Ulbrich, I. M., and Worsnop, D. R.: Real-time methods for estimating organic component mass concentrations from aerosol mass spectrometer data, Environmental Science & Technology, 45, 910–916, https://doi.org/10.1021/es102951k, 2011.

Srivastava, D., Favez, O., Petit, J.E., Zhang, Y., Sofowotee, U.M., Hopke, P.K., Bonnaire, N., Perraudin, E., Gros, V., Villenave, Albinet, A.: Speciation of organic fractions does matter for aerosol source apportionment. Part 3: Combining off-line and on-line measurements, Science of the Total Environment, 690, 944-955, https://doi.org/10.1016/j.scitotenv.2019.06.378, 2019.

Zhang, Q., Jimenez, J. L., Canagaratna, M. R., Ulbrich, I. M., Ng, N. L., Worsnop, D. R., and Sun, Y.: Understanding atmospheric organic aerosols via factor analysis of aerosol mass spectrometry: A review, Analytical and Bioanalytical Chemistry, 401, 3045–3067, https://doi.org/10.1007/s00216-011-5355-y, 2011.

*Reviewer 2:*

*General comment The paper is focused on characterization of organic aerosol in New Delhi (India) using an ACSM and some additional measurements. The approaches used with PMF receptor model allow to get some information about the sources and the trends in primary and secondary organics. The topic is interesting, up-to-date and suitable for the Journal. In general, it is not easy to read this paper because most of the figures and results are reported in the supplementary material (73 pages) that seems to be richer than the main paper. Authors should think about bringing some of the main results in the paper removing them from the supplementary material. His at least for the results that authors say to be important like the relationships with PBL height or the influence of the ventilation coefficient. Further, there are some aspects that are not clear (see my specific comments) that should be addressed in a revision step.*

Response:  We recognize that there is a lot of material included as supplemental information. This is necessary for justification and interpretation of the 12 PMF factor analyses presented in this work. We did not include the information in the main manuscript for readability. .

*Specific comments*
*Abstract. Thermodynamic modelling. What kind of modelling and for what?*

Response: We conducted phase equilibrium modeling of aerosols using the extended aerosol inorganics model (E-AIM) to estimate the gas phase concentrations, and to obtain total concentrations after combining it with measured PM concentrations.

In the revised manuscript, we have updated the text (Abstract):

"Phase equilibrium modelling of aerosols using the extended aerosol inorganics model (E-AIM) predicts equilibrium gas-phase concentrations and allows evaluation of the importance of ventilation coefficient (VC) and temperature in controlling primary and secondary organic aerosol."

*Abstract (and also in the main text). Authors speak of inter-annual variability, however, having only 15 months of measurements, the possibility to see an inter-annual trend is optimistic to say the least. I would suggest to change this aspect.*

Response: We agree with the reviewer that 15 months of data provides limited information on interannual variability. There are very few campaigns in India taking systematic year-on-year measurements of particle concentrations and composition for entire seasons at 1 minute time resolution. This makes the collected dataset detailed enough to capture interannual variability between the winters and springs of 2017 and 2018.

We have rephrased "interannual variability" in the manuscript as "interannual variability between the winters and springs of 2017 and 2018".

*Page 2 (line 8). Molecular markers such as...?*

We have updated the text to be more specific (Section 1):

"They attribute chloride to sources such as coal combustion and biomass and waste burning using molecular markers. For example, hopanes such as S and R homohopane isomers are tracers for coal combustion, PAHs such as phenanthrene and benzo(a)anthracene are tracers for coal and biomass burning, sugar anhydrosaccharides such as levoglucosan and mannosan are tracers for wood/biomass combustion (Pant et al., 2015). They also attribute higher winter concentrations to condensation of semivolatile ammonium nitrate and ammonium chloride during low-temperature conditions, weaker wind speeds, and shallow atmospheric boundary layer in the winter season (Pant et al., 2015, 2016a). In recent years, detailed source-specific profiles of combustion and non-combustion sources in the South East Asian region have been developed as a part of studies such as the NAMaSTE campaign; for example, to discern between garbage burning and dung

burning, tracers such as 1,3,5-Trimethylbenzene and coprostanol respectively have been identified (IIT Bombay, 2008, Stockwell et al, 2016, Goetz et al., 2018, Jayarathne et al., 2018)."

*Page 2 (lines 21-32). I agree that the high temporal resolution could furnish additional information compared to receptor models applied to 24h samples. However, in the case of using ACSM only some chemical species are available and there are limitations due to the missing information on metals and other compounds (for example levoglucosan and similar). This aspect should be clearly mentioned and a reference to the recent work of Belis et al (Atmospheric Environment 123 (2015) 240e250) regarding PMF and receptor models performances should be added.*

Response: While non-refractory molecules are heavily fragmented by the electron impact ionization within the ACSM, some ion fragments have been identified as tracer ions for specific parent molecules, such as those at m/z 60 and 73 for levoglucosan (Ng et al., 2011). The limited importance of other species such as metals contributing to $PM_1$ in Delhi is suggested by the strong correlations ($R^2$ of 0.85 and slope of 0.96) of the ACSM based $PM_1$ + BC concentrations with SMPS-$PM_1$ estimates. Details of these correlations can be found in Gani et al (2019). Additionally, as Gani et al (2019) notes, "most of the PM1 was composed of non-refractory material and BC was consistent with past literature from Delhi which observed that metals and other non-refractory crustal materials, which we did not measure in this study, constituted less than 5% of $PM_1$ (Jaiprakash et al., 2017)."

As per the reviewer's suggestion, we have updated the text (Section 2.1):

"PMF2 has been identified as an appropriate receptor modeling technique that can be deployed for quantifying source contributions for air quality management (Belis et al., 2015)".

In addition, we have updated the text elsewhere as well (Section 3):

"Thus, the source apportionment conducted was limited to separating primary and secondary OA in all seasons (and further, primary into BBOA and HOA in Spring 2018). Further separation of the factors may be  possible with high-resolution data combined with application of factor constraints (Aiken et al., 2009; Elser et al., 2016; Al-Naiema et al., 2018) and source-specific measurements such as metals/metal ions for combustion or biomass burning emissions (Jaiprakash et al, 2017), or α-methylglyceric acid for biogenic secondary OA (Srivastava et al, 2019)."

In addition, the updated manuscript includes a review of recent source apportionment studies on $PM_{2.5}$ in Delhi conducted using different techniques/data and how our results compare to the general conclusions of those previous studies:

In the introduction:

"Most receptor modeling studies have principally relied on a small number of daily or multi-day filter-based samples collected over temporally restricted sampling periods, thereby limiting the possible application of factor analysis techniques such as positive matrix factorization (PMF) to

quantify source contributions for entire seasons at the site. Further, despite Delhi being a continental site, multiple studies attribute significant portions of finer fractions of PM to a sea-salt origin (Sharma et al., 2014; Sharma and Mandal, 2017). Recent studies have also developed bottom-up emissions inventories for the National Capital Territory (NCT) region encompassing the city of Delhi (Guttikunda et al, 2013), and conducted multi-season multiple-site $PM_{2.5}$ source apportionment using bottom-up approaches (IIT Kanpur, 2016, ARAI and TERI, 2018). Generally, sources such as industrial emissions, transport emissions, residential combustion, power plant emissions, biomass and waste burning, and dust are shown to contribute substantially to $PM_{2.5}$ in all these studies. Although these studies accounted for primary organic carbon and secondary inorganic species such as sulfate and nitrate, they provide limited information regarding secondary organics."

In the conclusion:

"Because many bottom-up source apportionment efforts for Delhi are based on local (<50-100 km) scale inventories of primary emissions, they may neglect the important contributions of regionally transported primary and secondary aerosol from upwind regions. Here, we demonstrate that the submicron aerosol in Delhi experiences substantial atmospheric processing. Nevertheless, our results are in broad agreement with these inventories: 1) colder seasons are accompanied by higher concentrations and more diverse sources, including higher biomass burning emissions, 2) a significant aerosol mass fraction in Delhi is attributable to non-local sources, and 3) industries are a significant source of fine particulate matter (Guttikunda et al, 2013, IIT Kanpur, 2016, TERI, 2018). However, the lack of consistent emission inventories limits the feasibility of actions plans like GRAP, given that their implementation restricts economic activity (CEEW, 2019)."

*Section 2 (methods). ACSM is working on PM1 instead, other measurements have been done on PM2.5. Why not on the same size fraction? The differences should be explained. In addition, BC, and UVPM are not shown at all in the main paper but only in the supplementary material is this meaning that these species are not so important in the framework of the results?*

Response: The ACSM and AE33 deployed for the Delhi Aerosol Supersite (DAS) study are suitable for measuring $PM_1$ and $PM_{2.5}$ respectively. However, both instruments were on sampling lines with $PM_{2.5}$ cyclones (50% cutoff for aerosols 2.5 μm in diameter). Also, the sum of hourly averaged $PM_1$ concentration (measured by the ACSM) and BC concentrations (measured by the AE33) at this site are correlated to $PM_{2.5}$ measured at the nearest monitoring station operated by the Delhi Pollution Control Committee (DPCC), R.K. Puram (3 km away) with a slope of ~0.85, $R^2$~0.6 (Gani et al., 2019). This suggests that a $PM_1$ inlet likely captures the bulk of $PM_{2.5}$ aerosols.

At the time, the $PM_1$ inlet was standard for online aerosol mass spectrometers. Measurement of $PM_{2.5}$ requires the use of a different aerodynamic lens and vaporizer. Laboratory evaluation of the $PM_{2.5}$ system, published a few months prior to the beginning of our campaign, showed that the new vaporizer produced markedly different ion fragmentation patterns of aerosols (Xu et al., 2016). The consistency of fragmentation patterns is essential for factor identification using

reference profiles from mass spectra obtained across the world (Ng et al., 2011). Therefore, we used the standard $PM_1$ inlet with the ACSM.

As for BC and UVPM, we use BC as a primary aerosol tracer, with the Sandradewi model-based biomass burning component of black carbon ($BC_{BB}$) being used as one of the tracers for biomass burning. In addition, we use $\Delta C$, the difference between UVPM and BC, as a tracer for biomass burning. Therefore, neither BC nor UVPM were included in the PMF analysis directly. Details of the tracer selection can be found in Section 2.2 Factor Selection.

*Page 3 (section 2.1). Fifteen months divided into six seasons, why not using calendar seasons, I mean one year divided in four seasons?*

Response: Delhi's climate is not clearly defined by the four seasons that are commonly used in many other countries, as there are five distinct periods. In this work and Gani et al (2019), we have used the Indian National Science Academy's definition that classifies seasons based on temperature and climate. In effect, this definition divides one year into 5 seasons-winter, spring, summer, monsoon, and autumn.

Reference: Indian National Science Academy: Seasons of Delhi, https://www.insaindia.res.in/climate.php, 2018.

*Section 2.2. How it is used CO? In the PMF? Again, no trace of CO is reported in the main text.*

Response: In this study, carbon monoxide (CO) was not measured. Instead, when available, we used CO measured about 11 km (~7 miles) from our site at a fixed monitoring location RK Puram maintained by the Central Pollution Control Board (CPCB), Government of India, as an external traffic and combustion tracer.

We have updated the text (Section 2.2):

"In this study, carbon monoxide (CO) was not measured. When available, we used CO measured about 11 km (~7 miles) from our site at a fixed monitoring location RK Puram maintained by the Central Pollution Control Board (CPCB), Government of India, as an external traffic and combustion tracer."

*Page 5 (lines 26-28). It is reported that OOA correlates strongly with both sulphate and nitrate. However, in several sites nitrate and sulphate have different seasonal trends with sulphate larger in the warm seasons and nitrate larger in the cold seasons because of its thermal instability. This is true for both ammonium nitrate and sodium nitrate coming from aged marine salt. If I have well understood there is a correlation between SOA with both nitrate and sulphate suggesting that at this they have the same trend. Is this true? In case it will be useful to discuss this aspect mentioning explicitly the similarity in these trends that is not often observed at other sites.*

Response: In this study, we have conducted receptor modeling analysis using Positive Matrix Factorization (PMF) applied on organic mass spectra only and combined organic and inorganic mass spectra. While the OOA factor in PMF analysis conducted using organic mass spectra only

correlates with both nitrate and sulfate, in the combined organic and inorganic mass spectra based PMF analysis, OOA associated with nitrate and sulfate can be separated into OOA mixed with Ammonium Nitrate (AN-OOA) and OOA mixed with Ammonium Sulfate (AS-OOA) and the two components have different diurnal trends (Section 3.2.2, Fig. 6). Thus, the correlation of nitrate and sulfate with OOA in organic-only PMF analysis is likely a limitation of the technique when applied to the Delhi organic only MS dataset. As we have shown, this limitation can be addressed by explicitly including inorganic MS contributions in PMF analysis. This aspect has been discussed in detail in Sect. 3.2.2 and summarized in Sect. 4 Conclusions.

*Page 5 (line 30-31). Actually, looking at Figure 2 POA and OOA seems quite comparable. Are the difference mentioned statistically significant?*

Response: Since POA and OOA presented in Figure 2 are non-normally distributed (Gani et al., 2019) as is common for many environmental datasets, we conducted the two-tailed Kruskal-Wallis test and the Wilcoxon Signed Rank test. Results from both tests indicate that the overall distributions are significantly different ($p<0.05$). For detailed differences of summary statistics between POA and OOA, we refer the reviewer to Table 3.

*Page 7 (lines 9-10). This sentence is not correct because at night there is not a decreasing PBL, rather at sunset a new shallow boundary-layer is established generally thermally stable that could trap pollutants and will evolve at sunrise the day after mixing air from ground level with air masses above, see for example MeteorologischeZeitschrift, Vol. 21, No. 4, 385-398 (August 2012). I believe that authors should explain better this part.*

Response:

We have updated the text (Section 3.1):

"We observed larger nighttime (1800–2200 hours) and smaller daytime (0600–0900 hours) POA peaks. As the sun sets, radiative cooling of the ground surface flips the ambient temperature profile in the surface layer, with ambient temperature increasing with altitude. This inversion layer is thermally stable and traps pollutants at nighttime. As the sun rises, radiative heating warms up the ground surface and the ambient temperature profile returns to a decreasing trend with altitude, which is thermally unstable. Thus, the nighttime and daytime periods encounter similar thermal transition phases, albeit in opposite directions (Figs. 3, S15) (Seinfeld and Pandis, 2006). However, the nighttime period is accompanied by minimal photochemical conversion of POA to SOA in the evening, confirmed by the lower OOA, AN-OOA, and AS-OOA concentrations in the evening (Sect. 3.2). These dynamics could explain the larger nighttime and the smaller daytime POA peaks."

*Page 12 (lines 14-15). These percentages are really so different statistically?*

Response: We conducted the Kruskal-Wallis test based one-tailed non-parametric multiple comparison tests for the 6 seasonal fractional contributions of POA. Indeed, overall distribution

of spring 2018 is significantly higher compared to the fractions in other seasons ($p < 0.05$). However, spring 2017, winter 2017 and winter 2018 are statistically similar distributions.

We have updated the text (Section 3.3):

"Spring 2018 POA fraction varies between 34–75%, higher than the winters of 2017 and 2018 and spring 2017 (20–68%), and summer 2017 and monsoon 2017 (23–53%)".

*Page 12 (lines 18-19). The larger primary emissions during cold months is not reflected in the percentages above because the maximum appears to be during spring.*

Response: As shown in the previous response by the Kruskal-Wallis test, fractional contributions of POA in Spring 2017, with two factors only, are in line with the two winters. Spring 2018, on the other hand, is different from Spring 2017 due to the separation of an additional BBOA factor in that season. This separation likely resulted in the higher POA fraction in Spring 2018. Additionally, the Kruskal-Wallis test based one-tailed non-parametric multiple comparison tests show that temperatures in summer and monsoon are significantly higher than temperatures in springs which are significantly higher than those in the winters ($p < 0.05$). Therefore, spring and winter can be generally categorized as the colder months whereas summer and monsoon as the warmer. Thus, using the term "colder" months is reasonable.

*Figures 3b and 3c. The intercepts are missing, are they negligible?*

Response: The intercept in figure 3b is 0.001 and in figure 3c is 7e-04, negligible in comparison to the fractional contributions being represented in the figures.

*Figures 4, 5, 6, and 7. Please report in the caption what is the difference between continuous lines and marks.*

Response: The captions specify that lines correspond to means and marks correspond to medians in the figures.

References:

Aiken, A. C., Salcedo, D., Cubison, M. J., Huffman, J. A., DeCarlo, P. F., Ulbrich, I. M., Docherty, K. S., Sueper, D., Kimmel, J. R., Worsnop, D. R., Trimborn, A., Northway, M., Stone, E. A., Schauer, J. J., Volkamer, R. M., Fortner, E., de Foy, B., Wang, J., Laskin, A., Shutthanandan, V., Zheng, J., Zhang, R., Gaffney, J., Marley, N. A., Paredes-Miranda, G., Arnott, W. P., Molina, L. T., Sosa, G., and Jimenez, J. L.: Mexico City aerosol analysis during MILAGRO using high resolution aerosol mass spectrometry at the urban supersite (T0) — Part 1: Fine particle composition and organic source apportionment, Atmospheric Chemistry and Physics, 9, 6633–6653, https://doi.org/10.5194/acp-9-6633-2009, 2009.

Al-Naiema, I. M., Hettiyadura, A. P. S., Wallace, H. W., Sanchez, N. P., Madler, C. J., Cevik, B. K., Bui, A. A. T., Kettler, J., Griffin, R. J., and Stone, E. A.: Source apportionment of fine

particulate matter in Houston, Texas: insights to secondary organic aerosols, Atmospheric Chemistry and Physics, 18, 15 601–15 622, https://doi.org/10.5194/acp-18-15601-2018, 2018.

ARAI and TERI: Source apportionment of PM2.5 and PM10 of Delhi NCR for identification of major sources. Available at http://www.teriin.org/sites/default/files/2018-08/Report_SA_AQM-Delhi-NCR_0.pdf, 2018.

Belis, C.A., Karagulian, F., Amato, F., Almeida, M., Artaxo, P., Beddows, D.C.S., Bernardoni, V., Bove, M.C., Carbone, S., Cesari, D. and Contini, D., 2015. A new methodology to assess the performance and uncertainty of source apportionment models II: The results of two European intercomparison exercises. Atmospheric Environment, 123, 240-250, https://doi.org/10.1016/j.atmosenv.2015.10.068, 2015.

CEEW: What is Polluting Delhi's Air? Understanding Uncertainties in Emissions Inventory. New Delhi: Council on Energy, Environment and Water, https://www.ceew.in/publications/what-polluting-delhi%E2%80%99s-air, 2019.

Elser, M., Huang, R.-J., Wolf, R., Slowik, J. G., Wang, Q., Canonaco, F., Li, G., Bozzetti, C., Daellenbach, K. R., Huang, Y., Zhang, R., Li, Z., Cao, J., Baltensperger, U., El-Haddad, I., and Prévôt, A. S. H.: New insights into PM2:5 chemical composition and sources in two major cities in China during extreme haze events using aerosol mass spectrometry, Atmospheric Chemistry and Physics, 16, 3207–3225, https://doi.org/10.5194/acp-16-3207-2016, 2016.

Gani, S., Bhandari, S., Seraj, S., Wang, D. S., Patel, K., Soni, P., Arub, Z., Habib, G., Hildebrandt Ruiz, L., and Apte, J. S.: Submicron aerosol composition in the world's most polluted megacity: the Delhi Aerosol Supersite study, Atmospheric Chemistry and Physics, 19, 6843–6859, https://doi.org/10.5194/acp-19-6843-2019, 2019.

Goetz, J. D., Giordano, M. R., Stockwell, C. E., Christian, T. J., Maharjan, R., Adhikari, S., Bhave, P. V., Praveen, P. S., Panday, A. K., Jayarathne, T., Stone, E. A., Yokelson, R. J., and DeCarlo, P. F.: Speciated online $PM_1$ from South Asian combustion sources – Part 1: Fuel-based emission factors and size distributions, Atmospheric Chemistry and Physics, 18, 14 653–14 679, https://doi.org/10.5194/acp-18-14653-2018, https://www.atmos-chem-phys.net/18/14653/2018/, 2018.

Guttikunda, S.K. and Calori, G. A GIS based emissions inventory at 1 km× 1 km spatial resolution for air pollution analysis in Delhi, India. Atmospheric Environment, 67, 101-111, https://doi.org/10.1016/j.atmosenv.2012.10.040, 2013.

IIT Kanpur: Comprehensive study on air pollution and greenhouse gases (GHGs) in Delhi, A report submitted to the Govt. of NCT Delhi and DPCC Delhi, 2016.

Jaiprakash, Singhai, A., Habib, G., Raman, R. S., and Gupta, T.: Chemical characterization of PM1 aerosol in Delhi and source apportionment using positive matrix factorization, Environmental Science and Pollution Research, 24, 445–462, https://doi.org/10.1007/s11356-016-7708-8, 2017.

Jayarathne, T., Stockwell, C. E., Bhave, P. V., Praveen, P. S., Rathnayake, C. M., Islam, Md. R., Panday, A. K., Adhikari, S., Maharjan, R., Goetz, J. D., DeCarlo, P. F., Saikawa, E., Yokelson, R. J., and Stone, E. A.: Nepal Ambient Monitoring and Source Testing Experiment (NAMaSTE): emissions of particulate matter from wood- and dung-fueled cooking fires, garbage and crop residue burning, brick kilns, and other sources, Atmos. Chem. Phys., 18, 2259–2286, https://doi.org/10.5194/acp-18-2259-2018, 2018.

Ng, N. L., Canagaratna, M. R., Jimenez, J. L., Zhang, Q., Ulbrich, I. M., and Worsnop, D. R.: Real-time methods for estimating organic component mass concentrations from aerosol mass spectrometer data, Environmental Science & Technology, 45, 910–916, https://doi.org/10.1021/es102951k, 2011.

Pant, P., Shukla, A., Kohl, S.D., Chow, J.C., Watson, J.G. and Harrison, R.M. Characterization of ambient PM2. 5 at a pollution hotspot in New Delhi, India and inference of sources. Atmospheric Environment, 109, 178-189, https://doi.org/10.1016/j.atmosenv.2015.02.074, 2015.

Pant, P., Baker, S. J., Goel, R., Guttikunda, S., Goel, A., Shukla, A., and Harrison, R. M.: Analysis of size-segregated winter season aerosol data from New Delhi, India, Atmospheric Pollution Research, 7, 100–109, https://doi.org/10.1016/j.apr.2015.08.001, 2016.

Seinfeld, J. H. and Pandis, S. N.: Atmospheric Chemistry and Physics: From Air Pollution to Climate Change, Third Edition, Wiley, New Jersey, 2016.

Sharma, S. and Mandal, T.: Chemical composition of fine mode particulate matter ($PM_{2.5}$) in an urban area of Delhi, India and its source apportionment, Urban Climate, 21, 106–122, https://doi.org/10.1016/j.uclim.2017.05.009, 2017.

Sharma, S., Mandal, T., Saxena, M., Rashmi, Rohtash, Sharma, A., and Gautam, R.: Source apportionment of $PM_{10}$ by using positive matrix factorization at an urban site of Delhi, India, Urban Climate, 10, 656–670, https://doi.org/10.1016/j.uclim.2013.11.002, 2014.

Srivastava, D., Favez, O., Petit, J.E., Zhang, Y., Sofowotee, U.M., Hopke, P.K., Bonnaire, N., Perraudin, E., Gros, V., Villenave, Albinet, A.: Speciation of organic fractions does matter for aerosol source apportionment. Part 3: Combining off-line and on-line measurements, Science of the Total Environment, 690, 944-955, https://doi.org/10.1016/j.scitotenv.2019.06.378, 2019.

Stockwell, C. E., Christian, T. J., Goetz, J. D., Jayarathne, T., Bhave, P. V., Praveen, P. S., Adhikari, S., Maharjan, R., DeCarlo, P. F., Stone, E. A., Saikawa, E., Blake, D. R., Simpson, I. J., Yokelson, R. J., and Panday, A. K.: Nepal Ambient Monitoring and Source Testing Experiment (NAMaSTE): emissions of trace gases and light-absorbing carbon from wood and dung cooking fires, garbage and crop residue burning, brick kilns, and other sources, Atmos. Chem. Phys., 16, 11043–11081, https://doi.org/10.5194/acp-16-11043-2016, 2016.

Xu, W., Croteau, P., Williams, L., Canagaratna, M., Onasch, T., Cross, E., Zhang, X., Robinson, W., Worsnop, D. and Jayne, J. Laboratory characterization of an aerosol chemical speciation

monitor with PM2.5 measurement capability, Aerosol Science and Technology, 51, 69-83, https://doi.org/10.1080/02786826.2016.1241859, 2017.